# Lactoferrin Alleviates Lipopolysaccharide-Induced Infantile Intestinal Immune Barrier Damage by Regulating an ELAVL1-Related Signaling Pathway

**DOI:** 10.3390/ijms232213719

**Published:** 2022-11-08

**Authors:** Chaonan Li, Xinkui Liu, Zhihong Huang, Yiyan Zhai, Huiying Li, Jiarui Wu

**Affiliations:** 1Beijing Key Laboratory of Food Processing and Safety in Forestry, College of Biological Sciences and Technology, Beijing Forestry University, Beijing 100085, China; 2School of Chinese Pharmacy, Beijing University of Chinese Medicine, Beijing 102488, China

**Keywords:** lactoferrin (LF), lipopolysaccharide (LPS), intestinal immune barrier damage, blood immune cells, ELAVL1

## Abstract

As the most important intestinal mucosal barrier of the main body, the innate immune barrier in intestinal tract plays especially pivotal roles in the overall health conditions of infants and young children; therefore, how to strengthen the innate immune barrier is pivotal. A variety of bioactivities of lactoferrin (LF) has been widely proved, including alleviating enteritis and inhibiting colon cancer; however, the effects of LF on intestinal immune barrier in infants and young children are still unclear, and the specific mechanism on how LF inhibits infantile enteritis by regulating immune signaling pathways is unrevealed. In the present study, we firstly performed pharmacokinetic analyses of LF in mice intestinal tissues, stomach tissues and blood, through different administration methods, to confirm the metabolic method of LF in mammals. Then we constructed in Vitro and in Vivo infantile intestinal immune barrier damage models utilizing lipopolysaccharide (LPS), and evaluated the effects of LF in alleviating LPS-induced intestinal immune barrier damage. Next, the related immune molecular mechanism on how LF exerted protective effects was investigated, through RNA-seq analyses of the mouse primary intestinal epithelial cells, and the specific genes were analyzed and screened out. Finally, the genes and their related immune pathway were validated in mRNA and protein levels; the portions of special immune cells (CD4^+^ T cells and CD8^+^ T cells) were also detected to further support our experimental results. Pharmacokinetic analyses demonstrated that the integrity of LF could reach mice stomach and intestine after oral gavage within 12 h, and the proper administration of LF should be the oral route. LF was proven to down–regulate the expression levels of inflammatory cytokines in both the primary intestinal epithelial cells and mice blood, especially LF without iron (Apo-LF), indicating LF alleviated infantile intestinal immune barrier damage induced by LPS. And through RNA-seq analyses of the mouse primary intestinal epithelial cells treated with LPS and LF, *embryonic lethal abnormal vision Drosophila 1* (*ELAVL1*) was selected as one of the key genes, then the ELAVL1/PI3K/NF-κB pathway regulated by LF was verified to participate in the protection of infantile intestinal immune barrier damage in our study. Additionally, the ratio of blood CD4^+^/CD8^+^ T cells was significantly higher in the LF-treated mice than in the control mice, indicating that LF distinctly reinforced the overall immunity of infantile mice, further validating the strengthening bioactivity of LF on infantile intestinal immune barrier. In summary, LF was proven to alleviate LPS-induced intestinal immune barrier damage in young mice through regulating ELAVL1-related immune signaling pathways, which would expand current knowledge of the functions of bioactive proteins in foods within different research layers, as well as benefit preclinical and clinical researches in a long run.

## 1. Introduction

An inflammatory reaction results from a series of complex chemical signals released by immune cells, which is a proven protective mechanism of the innate immune system and necessary for physiological reactions. However, some studies have shown that inflammation during the early stages of life may lead to poor neurodevelopment [1,2]. Therefore, it is crucial to avoid the occurrence of inflammatory reactions during infancy. As the largest immune organ in the human body, the intestinal tract is the first line of defense against various pathogens. It has the most contact with the external environment and subsequently affects inflammatory reactions [3]; that is, intestinal health determines the growth and health of infants. The intestinal barrier is instrumental in preventing harmful substances or pathogenic microorganisms in the intestinal cavity from entering the human body. The intestinal mucosal barrier represents the primary section of the intestinal barrier [4] and includes the intestinal mucosal epithelial cells. The close connection between the cells and the mechanical barrier comprises the bacterial membrane [5]. The chemical barrier comprises mucus secreted by the intestinal mucosal epithelial cells, digestive juice, and bacteriostatic substances produced by normal parasitic bacteria in the intestinal cavity [6]. The biological barrier consists of the micro-ecosystem formed by normal flora and the host in the intestinal tract [7]. The immune barrier is crucial and comprises intestinal mucosal lymphoid tissue, intestinal mucosal immune cells, and intestinal endocrine immunoglobulin A (slgA) [8]. Since infants and young children are in the process of growth and development, they are quickly affected by harmful antigens and diseases if lacking sufficient immune protection. Therefore, the innate immune barrier in the intestinal tract plays an especially pivotal role in maintaining their overall health.

Lipopolysaccharides (LPS), cell wall components in Gram-negative bacteria (such as *Escherichia coli*, *Salmonella*, and *Pseudomonas aeruginosa*) [9], are usually present in inferior milk products [10] and cause intestinal and systemic inflammatory reactions [11,12]. Studies have shown that LPS damages the intestinal mucosal barrier and increases intestinal permeability [11,13]. When the critical immune barrier is destroyed, multiple symbiotic microorganisms begin to extravasate, causing various complications and even systemic infections [14,15]. Since the gastrointestinal immune systems of newborns are fragile and immature, the LPS from Gram-negative bacteria in inferior milk products may result in excessive inflammation and potential injury [16].

Lactoferrin (LF) is a natural bioactive protein mainly derived from dairy products. It was first discovered in 1939 [17] and was successfully isolated and purified from human milk and milk in 1960 [18]. Because this isolated protein is structurally similar to serum transferrin and can reversibly bind with iron (Fe^3+^), it is classified as a member of the transferrin family [19]. The unique structural characteristics of LF present a variety of nutritional and medicinal advantages. LF can transport Fe^3+^ and fight free radicals in the body [20,21] while also exhibiting antibacterial and antiviral activity. It regulates immunity, provides intestinal barrier protection, and resists Gram-positive and Gram-negative bacteria [22,23]. For example, LF can bind with the lipoproteins in the outer bacterial membrane to form receptor complexes, consequently controlling the interaction between bacteria and host cells [24,25]. The antiviral effect of LF is mainly reflected during the early stage of viral infection, preventing the viral infection of host cells and inhibiting the reproduction of viruses after host cell infection [26,27]. Interestingly, studies have shown that a higher LF content in breast milk can improve the overall immunity of infants, protecting them from bacterial infection and inflammation [28,29]. In Vivo and in Vitro experiments demonstrate that LF provides intestinal barrier protection by restoring the tight junction (TJ) morphometry, blocking the cleavage of caspase-3, and resuming the drop in transepithelial resistance (TER) in inflammatory bowel disease (IBD) models [30,31,32]. In addition, LF can enhance intestinal epithelial cell proliferation, regulate cytokines (reducing the secretion and expression of TNF-α, IL-8, IL-6, and NF-κB), and enhance immune cell function (activating CD4^+^ T cells in the colon; Treg cell differentiation encourages dendritic cells and macrophages to become tolerogenic phenotypes) to reduce inflammatory reactions and maintain an immune steady–state [33,34,35,36]. However, the effect of LF on the intestinal immune barrier in infants and young children remains unclear; the specific mechanism on how LF inhibit infantile enteritis by regulating immune signaling pathways is still unrevealed. Therefore, this study constructed in Vitro and in Vivo intestinal immune barrier damage models utilizing LPS, evaluated the ability of LF to alleviate LPS–induced intestinal immune barrier damage, and investigated the related molecular mechanism.

## 2. Results

### 2.1. Pharmacokinetic Study of LF by UPLC

Mass spectrometry detection showed that the AU of the LF sharply rose dramatically around 2.3 min, reaching a peak at 2.41 min, followed by a sharp decline to 0 in 0.2 min (Figure 1A). The LF pharmacokinetics were evaluated in the blood, stomach contents, and intestinal contents of the young mice after oral gavage or intraperitoneal injection. The LF concentration peaked at 2 h, followed by a slight decrease, reaching nadir after 12 h. The peak LF concentration was in the order of intestinal tissues > stomach tissues > blood (Figure 1B), indicating that the bioavailability of LF through oral gavage was higher than the one through I.P., and the majority of LF directly reached into intestinal tissue through oral gavage. Additionally, we calculated several pharmacokinetics parameters of LF in the present model; as through oral gavage, the peak concentrations (Cmax) of LF in blood, intestinal tissue and stomach tissue were 208 mg/L, 1905 mg/L, and 863 mg/L, respectively. Through I.P. injection, the Cmax of LF in blood, intestinal tissue and stomach tissue were 219 mg/L, 23 mg/L and 60 mg/L, respectively. The half-life (t1/2) in blood samples was 3.75 h (by oral gavage) and 1.87 h (by I.P. injection).

### 2.2. The Influence of LF with Different Fe^3+^ Saturation Levels on Iinflammatory Factors In Vitro and In Vivo

The ELISA results showed that LPS at doses exceeding 10 µg/mL triggered dose-dependent intracellular inflammatory responses in the primary intestinal epithelial cells of the mice. This significantly decreased the cell survival rate (Figure 2A) and increased the expression of IL-1β, IL-6, TNF-α and IFN-γ compared to the cells without LPS treatment (* *p* < 0.05, Figure 2B). Furthermore, 2 h of LF (both holo- and apo-transporters) treatment significantly elevated the survival rate of the LPS-induced primary intestinal epithelial cells of the mice compared to the LPS-induced cells without LF treatment (# *p* < 0.05, Figure 2A), and inhibited the expression of intracellular inflammatory factors (# *p* < 0.05, Figure 2B). Therefore, the protective effect of apo-LF against LPS-induced intestinal injury was better than holo-LF (Figure 2B,C). Moreover, the expression of the inflammatory factors in the serum of the young mice was determined via ELISA after 14 d of drug administration. The results showed that the expression of IL-1β, IL-6, TNF-α, and IFN-γ in the serum of LPS-induced young mice was significantly higher than that in the normal mice, while the LF + LPS group exhibited a remarkably lower expression in serum IL-1β, IL-6, TNF-α, and IFN-γ than the LPS group (Figure 2C). Moreover, the expression levels of the serum inflammatory factors in the Apo-LF + LPS group were slightly lower than that in the Holo-LF + LPS group (Figure 2C). These results demonstrated that the LF with different transferrin saturation levels exhibited a protective effect against acute LPS-induced intestinal injury, while the activity of Fe^3+^-free LF was stronger than that of Fe^3+^-saturated LF. Thus, Fe^3+^-free LF was used for all subsequent experiments in the present study.

### 2.3. RNA-Seq Analyses of the LF in the Primary LPS-Induced Primary Intestinal Epithelial Cells of the Mice

To further investigate the LF mechanism in intestinal injury, RNA-seq analysis was performed on normal primary intestinal epithelial cells, LPS-induced cells, LF-treated normal cells, and LF-treated LPS-induced cells derived from the mice. The results showed a higher inter-group correlation and a relatively lower within-group correlation. The samples in the control group especially could be clearly distinguished from the other three groups (Figure 3A,B). The mRNA expression of the immune- and inflammation-related genes (*Il1r1, Vegfd, Cebpb, Fos, Elavl1, Ppard, Olr1, Cxcl5, Cxcl1, Lif, Lbp, Il6, Map3k8, Cd14, Cxcl3, Cxcl10, Cxcl2, Mmp3, Lck*, and *Ccl4*) was substantially up-regulated in the primary LPS-induced mouse intestinal epithelial cells compared to the normal cells (Figure 3C). The addition of LF significantly decreased the expression of these genes in the LPS-induced cells (Figure 3C). GSVA analysis revealed that LF intervention significantly changed the activities of multiple signaling pathways, especially PI3K AKT mTOR signaling (Figure 3D,E). Compared to the primary LPS-induced mouse intestinal epithelial cells, the relative activity of PI3K/AKT/mTOR signaling in the LPS-induced cells treated with LF was significantly reduced (Figure 3F). Therefore, ELAVL1 was primarily selected as the candidate factor and PI3K/AKT/mTOR pathway was selected as the key pathway in the present injury model, which might be regulated by LF.

### 2.4. The Effect of LF on the mRNA and Protein Expression of Immune Indicators In Vitro and In Vivo

To investigate the immunomodulatory effect of LF on intestinal injury, the mRNA and protein expression of the immune-related molecules in the primary LPS-induced intestinal mouse epithelial cells and the blood of LPS-induced young mouse colitis models were determined after LF treatment. The immune-related molecules in the LPS-induced cells were significantly enhanced compared to the mouse primary intestinal epithelial cells. Contrarily, the mRNA and protein expression of these genes in the LPS-induced cells treated with LF was remarkably lower than in the LPS-induced cells. Compared to the normal mice, the immune-related molecules in the blood of the mice with colitis were significantly enhanced. In contrast, the mRNA and protein expression of these genes in the blood of the LF-treated mice with colitis was substantially lower than that in the mice with colitis, indicating that LF posed an immunomodulatory effect on intestinal injury through regulating the expression levels of immune-related factors.

### 2.5. The Immunomodulatory Functions of LF In Vivo

To further explore the immune responses exerted by LF in Vivo, flow cytometry was used to analyze the blood T cell populations in the LF-treated mice via oral gavage. The results demonstrated that the ratio of blood CD4^+^/CD8^+^ T cells was significantly higher in the LF-treated mice than in the control group (Figure 4A–C). These findings indicated that LF distinctly reinforced the immunity of young mice, further validating the overall anti-inflammatory bioactivity of LF [37,38]. Moreover, in mRNA expression level of the LPS-treated mice, the levels of *ELAVL1*, *PI3K*, *NF-κB*, *TNF-α*, and *IL-1β* were significantly higher than those in the control and LF groups (Figure 5A). Accordingly, an increase in the protein expression levels of these genes was also observed in the LPS-induced mice (Figure 5B,C). The mRNA and protein expression levels of ELAVL1, PI3K, NF-κB, TNF-α, and IL-1β in the LPS-induced mice treated with LF were significantly lower than those in the LPS-induced mice without LF, further verifying that LF could inhibit intestinal injury through regulating the expression levels of ELAVL1, PI3K, NF-κB, TNF-α, and IL-1β (Figure 5A–C).

## 3. Discussion

The intestinal barrier is critical for maintaining intestinal homeostasis by separating the internal milieu from the external environment, while the disruption of the intestinal barrier is closely related to various diseases [39]. The intestinal barrier is indispensable in early life to prevent infection, inflammation, and food allergies [40]. It consists of four functional barriers: a mechanical, a chemical, an immunological, and a biological barrier. As a guardian of intestinal integrity and the entire organism, the intestinal immunological barrier has evolved tightly regulated mechanisms to inhibit unnecessary inflammatory responses while still protecting the organism from pathogens [41]. The intestinal immune system is composed of gut-associated lymphoid tissue (GALT), which is divided into dispersed and organized lymphoid tissue, including intestinal intraepithelial lymphocytes (iIEL), lamina propria lymphocytes (LPL), Peyer’s patches (PP), mesenteric lymph nodes, and intestinal plasma cell secretory immunoglobulin A (sIgA) [42]. The lymphocytes in the intestinal immune barrier can be activated to produce inflammatory mediators and regulate immune responses. If an immune response is not controlled in time, intestinal homeostasis may be disrupted, and the integrity of the barrier may be destroyed, leading to the occurrence of disease [43,44,45,46,47]. Immune derangements increase barrier permeability, and the LPS produced by bacteria are more likely to pass through and enter the intestine, leading to chronic inflammation. Extended intestinal inflammation can also cause low-grade inflammation in other extraintestinal organs, even systemic inflammation [48,49,50,51,52]. Restoring epithelial barrier function plays a crucial role in shaping gut mucosal homeostasis and suppressing inflammatory bowel disease (IBD) [53,54]. Research has proven that neutrophils in intestinal immune barrier could modulate pathogenic and repair processes of IBD [55]. IBD is more common in infants than in older children due to immature intestinal barriers, leading to malnutrition and weight loss [56,57,58]. A study found multiple, longitudinally sampled immune system/inflammatory biomarkers in young infants, which characterized chronic inflammation in this population [59]. However, the pathogenesis of IBD remains unclear, while the activation of inflammation is related to natural and adaptive immunity. These pro-inflammatory cytokines may damage intestinal barrier function and permeability [60,61].

LF is a kind of Fe^3+^-binding glycoprotein, which mainly exists in milk and belongs to the transferrin family along with serum transferrin, egg transferrin, melanin transferrin, and carbonic anhydrase inhibitors [62,63]. LF is an important part of natural immunity and plays a crucial role in host defense against infection and inflammation, which is achieved by regulating natural and adaptive immunity [64,65,66,67]. Some studies have shown that LF can decrease the content of inflammatory factors and the expression of related genes in an LPS-induced enteritis model, reduce oxidative stress, and maintain intestinal barrier integrity [68,69,70,71,72]. LF can also alleviate intestinal injury by regulating the immune mechanism. Furthermore, LF can increase the proportion of natural killer (NK) cells in mice and humans, enhance phagocytosis, regulate myelopoiesis, and induce T cell proliferation and maturation [73,74,75,76,77]. Lutaty et al. showed that the bioactive tripeptide phenylalanine-lysine-aspartic acid and phenylalanine-lysine-glutamic acid in LF fragments induced macrophages to reprogram into anti-inflammatory/pro-decomposition phenotypes. This inhibited the LPS-activated pro-inflammatory signal transduction and cytokine production of macrophages and enhanced inflammation regression [78]. A study by Zong indicated that LF maintained the balance of CD3^+^ and CD8^+^ T cells, B cells, and NK cells to defend against LPS-induced intestinal inflammation in mice [72]. Moreover, dietary LF used in a study could improve the growth performance and the overall immunity of Sobaity seabream (Sparidentex hasta) juveniles, mainly through regulating the expressions of immune-related and growth-related genes [79]. All the above studies indicated that lactoferrin could regulate immune system through affecting multiple signal pathways, however, the role of the ELAVL1-related pathway in intestinal immune barrier injury models was still unrevealed, especially in infants and young children models. Thus, in some degree, the present manuscript expands current knowledge of the functions of bioactive proteins from foods within different research layers, such as the medical effects of lactoferrin with different iron saturations, on alleviating infantile enteritis. Meanwhile, we uncovered the molecular mechanism of an ELAVL1-induced signaling pathway regulated by LF, which would likely benefit preclinical and clinical researches in a long run.

Moreover, the iron saturation of lactoferrin determines the bioactivities of LF, which was verified in several studies. Fan verified that the anti-inflammatory effect of Apo-LF was stronger than the one of Holo-LF in a young mice enteritis model, which could be elucidated by the iron saturation of LF [80]. El-Nasr proved that combined ferrous sulphate and lactoferrin was more effective than ferrous sulfate in pregnant women with IDA, with fewer gastrointestinal adverse effects and better effect on neonatal iron store, indicating that iron saturation balance in the combination of ferrous sulphate and LF might be helpful for the health of pregnant women [81]. Another research showed that the iron binding promoted changes on tertiary structure of bLf and increased its structural stability, further influencing the bioactive effect of LF [82]. In our study, we compared LF with different iron saturations (Apo-LF or Holo-LF) in protecting mouse primary intestinal epithelial cells and young mice from LPS-induced inflammatory injury, and verified that LF without iron demonstrated stronger bioactivity in protecting infantile intestinal immune barrier. Therefore, the present study not only provides a theoretical basis for preclinical and clinical research in a long run, but also benefits the research and development of infant dairy products. Additionally, we performed the pharmacokinetics detection of LF in blood, intestinal tissue and stomach tissue, through oral gavage and I.P. injection, in order to compare the metabolic characteristics of LF in blood, intestines and stomach. We found that the majority of LF could directly reached into intestinal tissue and reserve its primary intact form. As through oral gavage, the peak concentration (Cmax) of LF in intestinal tissue was higher than the ones in blood and stomach tissue, verifying the bioavailability of LF through oral gavage was much higher than the one through I.P. injection. Through comparing the half-life (t1/2) of LF in blood samples, we found that the values of 3.75 h (by oral gavage) and 1.87 h (by I.P. injection) were shorter than our expected ones, further suggesting that how to prolong the metabolic course of LF was necessary.

## 4. Materials and Methods

### 4.1. Reagents

Two types of LF from bovine milk (Apo-LF and Holo-LF, purified >95%) were purchased from Sigma-Aldrich (St. Louis, MO, USA). Solarbio Technology Co., Ltd. (Beijing, China) provided the inflammatory factor detection kits. Analytical-grade chemicals were used in all sample pretreatment procedures. Chromatography grade acetonitrile (ACN) and trifluoroacetic acid (TFA, JK Scientific, Beijing, China) were used for Ultra Performance Liquid Chromatography (UPLC). Ultrapure water was prepared using a Milli-Q purification system (Millipore, Bedford, MA, USA), while the primers for the reverse transcription-polymerase chain reaction (RT-PCR) assay were synthesized by Sangon Biotech (Sangon, Shanghai, China). The primary antibodies, including PI3K, AKT, mTOR, p-PI3K, p-AKT, p-mTOR, and β-actin, as well as the secondary antibodies, were purchased from Abcam (Abcam, Shanghai, China), while the related reagents were purchased from Solarbio (Solarbio, Beijing, China). The enzyme-linked immunosorbent assay (ELISA) kits for IL-1β and TNF-α were bought from Abcam (Abcam, Shanghai, China).

### 4.2. Animals

For LF detection, 144 young mice (14–16 d old) were purchased from Charles River (Beijing, China) and divided into six groups, consisting of three oral administration groups and three intraperitoneal injection (I.P.) groups (*n* = 24). The oral dose and the I.P. dose was the same one, 1 g/kg body wight (b.w.). Eight time points were established in each group, namely 0 h, 0.5 h, 1 h, 2 h, 3 h, 6 h, 12 h, and 24 h, with three mice in each timepoint group (*n* = 3). At each time point, the intestinal tissues and the stomach tissues were gathered, the tissues (50 mg per mice) were cut into small pieces, 2 mL of normal saline was added into each sample, and the samples were put into the homogenizer to crush the homogenate, once every 30 s, and they were homogenized twice, with an interval of 30 s between the two times to avoid temperature rise. Then, the homogenates were centrifuged at 1500 rpm for 10 min, the supernatant samples were collected for further detection. Finally, the LF concentration in the intestinal tissues, stomach tissues, and blood of the mice was detected after oral gavage or intraperitoneal injection.

For primary cell separation, ten young mice (14–16 d old) were purchased from Charles River (Beijing, China) and executed via decapitation. About 2 cm of the intestinal tissue of each mouse was dissected and cut into small pieces (1 cm × 1 cm), which were digested in trypsin for cell separation and culturing.

For blood immune cell detection, ten young mice (14–16 d old) were purchased from Charles River (Beijing, China) and divided into two groups: control without any treatment, and LF group treated with oral gavage (250 mg/kg b.w.). Then, 24 h later, the mice were executed and blood samples were collected from the eyeballs. About 0.15 mL of blood was collected from each mouse and prepared for flow cytometry analysis of the immune cells.

All the animal experiments were approved by the Ethics Committee of the Sinoresearch Biotechnology Co., Ltd. (Beijing, China; permission number: ZYZC202207006S), conforming to internationally accepted principles for the care and use of experimental animals (NRC, 2011). All surgical procedures were performed under sodium pentobarbital anesthesia, and all efforts were made to minimize the suffering of the mice.

### 4.3. Detection of LF via UPLC

Blood and tissue samples were frozen until analysis. The frozen samples were placed in a water bath at 40 °C and stirred carefully to avoid frothing, after which they were precipitated using phosphate-buffered solution (pH = 7.5) and centrifugated at 5000 r/min for 15 min to obtain the supernatant. The solution was filtrated using 25 mm filters and 0.45 μm Cellulose Acetate Blue Luer-Lock filters (Restek, Bellefonte, PA, USA) and analyzed via UPLC. All samples were tested in triplicate.

As Chen et al. described, the chromatographic analysis was performed using a Waters Acquity UPLC system (Waters, Milford, MA, USA) equipped with a Waters 2998 Photodiode Array Detector and an Xbridge Protein BEH C4 column (100 mm × 2.1 mm, 1.7 µm partial size) maintained in a column oven at 60 °C and an injection volume of 10 μL [83]. A gradient program was applied, with the mobile phase consisting of a combination of solvent A (1% TFA in ultrapure water) and solvent B (1% TFA in ACN), which included 95% A and 5% B (initial), 62% A and 38% B (1.5–2.5 min), 40% A and 60% B (3–4 min), and 95% A and 5% B (4.1–6 min) at a flow rate of 0.5 mL/min, while the analytes were detected at 280 nm. The LF concentration in each sample was calculated according to the standard curve [83].

### 4.4. Separation and Culturing of the Primary Intestinal Epithelial Cells

After an adaptation period of 3 d, the young mice were executed, after which the intestinal tissue was excised and washed three times with ice-cold PBS buffer. Then, the tissue was cut into small pieces of approximately 1 mm^3^ and immersed in a DMEM/F12 medium containing a penicillin-streptomycin solution (100×) for 1 h. The supernatant was collected and mixed with collagenase XI (450 U/mL) and neutral protease (0.2 mg/mL) for digestion at 37 °C for 20 min, after which the sample was centrifuged (1500 g, 15 min), and the precipitate was saved. The cells were seeded into a 6-well plate, which was pre-coated with collagenase I (450 U/mL). After most of the cells were attached to the wells, the small intestinal epithelial cell colonies were screened, and the fibroblasts were discarded by washing and digesting several times. Finally, basically all the intestinal epithelial cells appeared in the microscopic field of vision, which could be utilized for the further experiments.

### 4.5. Cell Viability Detection Using a Cell-Counting Kit-8 (CCK-8)

The primary intestinal epithelial cells were seeded into a 96-well plate and cultured for about 24 h. The cells were first exposed to apo-LF or Holo-LF (10 g/L) for 8 h, after which LPS (a final concentration of 0–100 mg/L) was added to the wells and co-cultured for 48 h. A cell-counting kit-8 (CCK-8) was applied to measure cell viability, and the appropriate concentrations of two types of LF were ultimately selected to investigate the protective effect of LF.

### 4.6. Inflammatory Factors Detection via ELISA

ELISA was applied to detect the concentrations of several inflammatory factors in the mouse blood samples, including IL-1β, IL-6, TNF-α, and INF-γ. According to the protocols, a mixture of biotinylated detection antibodies was added to 300 μL mouse serum in each well, followed by the captured antibodies, and incubated for 4 h. Then, the wells were washed three times, and the bound inflammatory factors were measured at 450 nm using a microplate reader.

### 4.7. RNA-Seq Analyses

RNA-seq was performed using the primary intestinal epithelial cells of the mice in the four groups (control, LPS, LF, and LF-LPS), with three samples per group (*n* = 3). The RNA integrity was determined via agarose gel electrophoresis, while the RNA purity was measured using a NanoPhotometer spectrophotometer. The RNA-seq library was prepared using an Illumina library preparation kit according to the protocol of the manufacturer. The libraries were quantified with a Qubit 2.0 Fluorometer and qualified using an Agilent 2100 Bioanalyzer. Finally, the strand-specific cDNA libraries were sequenced with an Illumina NovaSeq 6000 by Gene Denovo Biotechnology Co., Ltd., China (Guangzhou, China). The quality of the raw sequence reads was evaluated with fastp (version v0.18.0) [84], while low-quality bases and adapter sequences were discarded. The trimmed reads were aligned with the mouse reference genome using HISAT2 (version v2.1.0) [85]. The transcripts were reconstructed and quantified with StringTie (version v1.3.4) [86]. The hallmark gene sets [87] obtained from the Molecular Signatures Database (MSigDB, version 7.4) were used to perform Gene Set Variation Analysis (GSVA) [88], while a linear model in the limma package [89] was applied to evaluate the significance of differences.

### 4.8. Flow Cytometry Analysis of Blood Immune Cells

Blood samples from the young mice in the control and Apo-LF groups were used to quantify the immune cells via flow cytometry. Before flow cytometry analysis, the blood cells were lysed with 1 × red blood cell lysis buffer (BioLegend, San Diego, CA, USA) and washed with ice-cold PBS. For cell surface staining, the cells were incubated with surface antibodies in flow cytometry buffer at 4 °C for 30 min. The subsequent PE-Cy7 anti-mouse CD45 (BioLegend, San Diego, CA, USA), FITC anti-mouse CD3 (BioLegend, San Diego, CA, USA), BV510 anti-mouse CD8a (BioLegend, San Diego, CA, USA), and PerCP/Cy5.5 antimouse CD4 (BioLegend, San Diego, CA, USA) antibodies were used for cell staining. The cells were collected in a BD FACS-Verse flow cytometer (BD, San Jose, CA, USA), and the data were analyzed using FlowJo software.

### 4.9. RT-PCR

The total RNA of the cells was extracted using Trizol. Here, 0.2 mL chloroform was added and mixed carefully for 20 s, after which the mixture was centrifuged for 12 min at 4 °C and 12,000 g. The upper sample was collected and transferred to a new tube, after which 0.5 mL isopropanol was added and mixed thoroughly. The sample was left to stand at 25 °C for 12 min and centrifuged for 12 min (4 °C, 12,000 g), after which the supernatant was carefully removed. Next, 1 mL of 75% ethanol-precipitated RNA was added, mixed thoroughly, and centrifuged for 12 min (4 °C, 10,000 g), and the supernatant was absorbed. Then, 60 µL of DEPC water was added to the tube to dissolve the purified RNA sample. The 2^−∆∆CT^ method was applied to quantify the target gene levels, which were normalized to the expression level of the internal reference.

### 4.10. Western Blotting

A protein extraction kit was applied to extract the total proteins from the cells. The samples were heated at 95 °C for 15 min and loaded into 12% SDS-polyacrylamide gel for electrophoresis. The proteins were then transferred onto a nitrocellulose membrane using a trans-blot kit and blocked in 5% BSA buffer for 1.5 h. Then, the primary antibodies and internal reference β-actin were added to the membrane and cultured for 2 h, respectively. The membrane was washed three times with TBST buffer, after which the secondary antibody was added and co-incubated for 60 min. Finally, the target proteins on the membrane were measured using an enhanced chemiluminescence reagent, while the protein bands were scanned and quantified with Quantity One software.

### 4.11. Statistical Analysis

All quantitative data used in the current study are presented as mean ± standard deviation. The statistical analysis of the experimental data was performed with GraphPad Prism version 6.0 (GraphPad Software, San Diego, CA, USA). Statistical comparisons were made via a Student’s *t*-test or one-way analysis of variance (ANOVA). The difference between groups was considered statistically significant at *p* < 0.05.

## 5. Conclusions

In summary, the present study simulates the ability of the LF in dairy products to alleviate intestinal injury in infants by demonstrating that it inhibits LPS-induced IBD in juvenile mice with impaired intestinal immune barriers. Transcriptome analysis results confirm that LF protects the intestinal immune barriers of infants by regulating the immune-related signaling pathways. Furthermore, the undegradable LF content in the blood, stomachs, and intestines of the mice is determined via oral administration and intraperitoneal injection. The pharmacokinetic parameters of the LF in the mice are obtained for the first time, and the results suggest that oral LF administration is more persistent than intraperitoneal injection. Therefore, oral LF administration is suggested for future clinical studies.

## Figures and Tables

**Figure 1 ijms-23-13719-f001:**
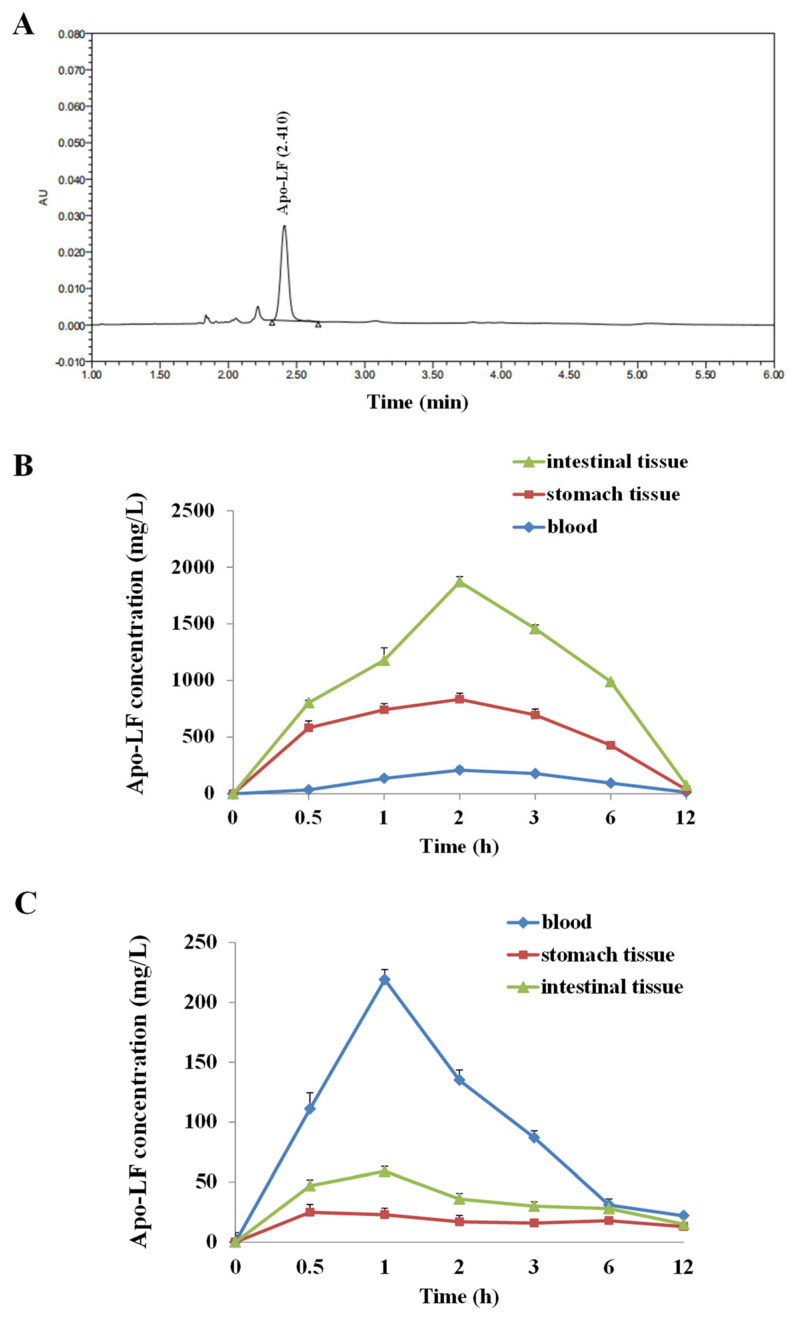
Pharmacokinetic analyses of Apo-LF. (**A**) Chromatogram of Apo-LF standard in testing solution from milk samples; (**B**) The concentration of Apo-LF in the intestinal tissues, stomach tissues and blood of mice after oral gavage; (**C**) The concentration of Apo-LF in the intestinal tissues, stomach tissues and blood of mice after intraperitoneal injection.

**Figure 2 ijms-23-13719-f002:**
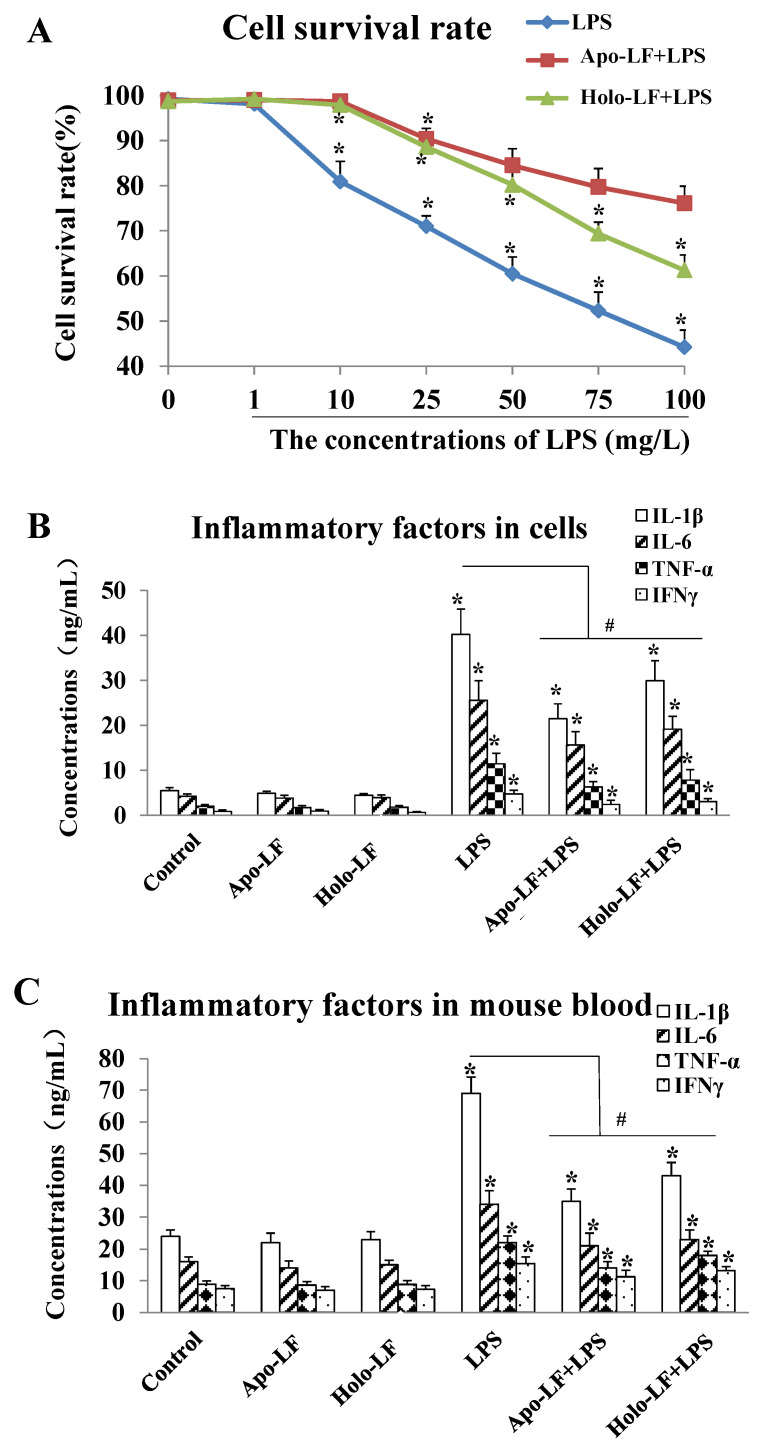
Lactoferrin with different iron saturations (Apo-LF or Holo-LF) protects mouse primary intestinal epithelial cells and young mice from LPS-induced inflammatory injury. (**A**) The cell survival rate of the LPS-induced mouse primary intestinal epithelial cells treated with saline, Apo-LF and Holo-LF; (**B**) The expression levels of inflammatory factors in the mouse primary intestinal epithelial cells with different interventions; (**C**) The expression levels of inflammatory factors in the serum of the young mice with different interventions. *n* = 3 mice per group. Data represent mean ± SD. * *p* < 0.05 by Student’s *t* test and ^#^
*p* < 0.05 by one-way ANOVA analysis.

**Figure 3 ijms-23-13719-f003:**
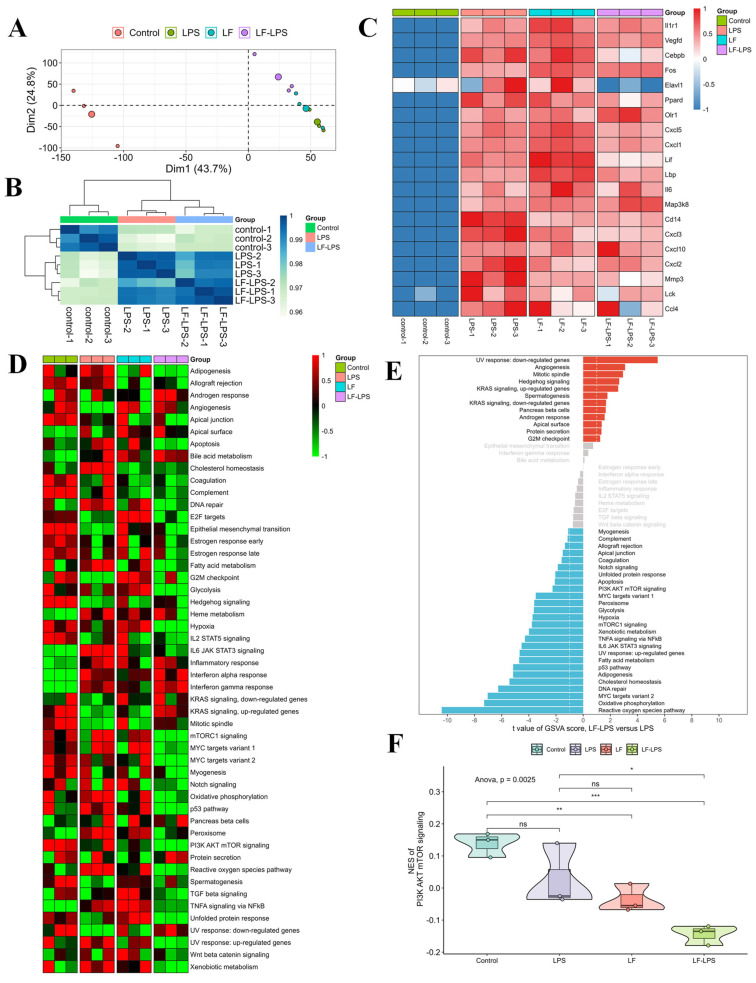
RNA-seq analyses of the mouse primary intestinal epithelial cells with different treatments. (**A**) PCA for the 12 samples; (**B**) Correlation heatmap of the samples in the control, LPS and LF-LPS groups; (**C**) Heatmap showing the expression of representative genes in the four groups; (**D**) Heatmap visualizing the activities of hallmark gene sets among the four groups; (**E**) Differential activities of hallmark gene sets in the LF-LPS versus LPS groups; (**F**) Difference in the activities of PI3K AKT mTOR signaling scored per sample by GSVA. (* *p* < 0.05, ** *p* < 0.01, *** *p* < 0.001, ns = no significant difference).

**Figure 4 ijms-23-13719-f004:**
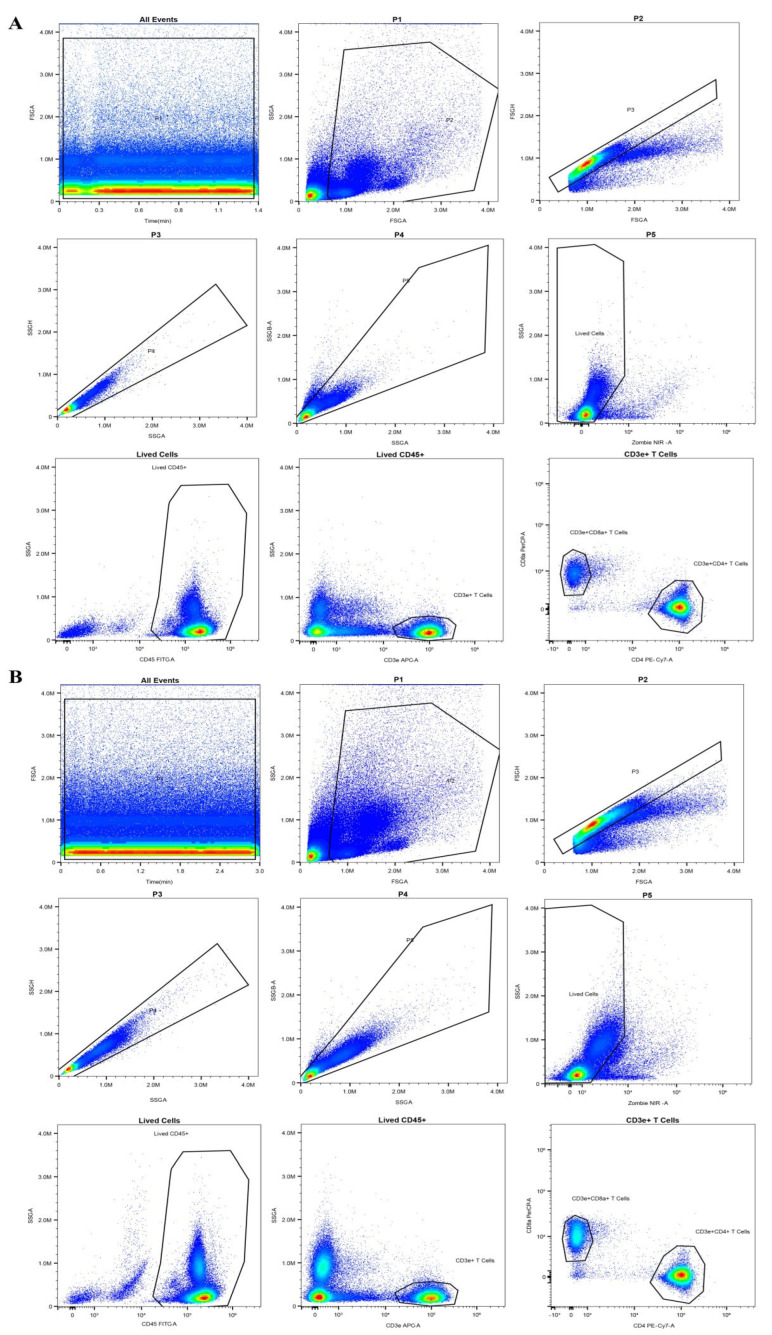
Effects of Apo-LF on immunocytes in blood of mice. (**A**) Representative flow cytometry analysis plot for blood CD4^+^ and CD8^+^ T cells in the mice from the control group; (**B**) Representative flow cytometry analysis plot for blood CD4^+^ and CD8^+^ T cells in the mice treated with lactoferrin (250 mg·kg^−1^ b.w.); (**C**) The difference in the ratio of blood CD4^+^/CD8^+^ T cells between the mice treated with saline and Apo-LF. *n* = 5 mice per group.

**Figure 5 ijms-23-13719-f005:**
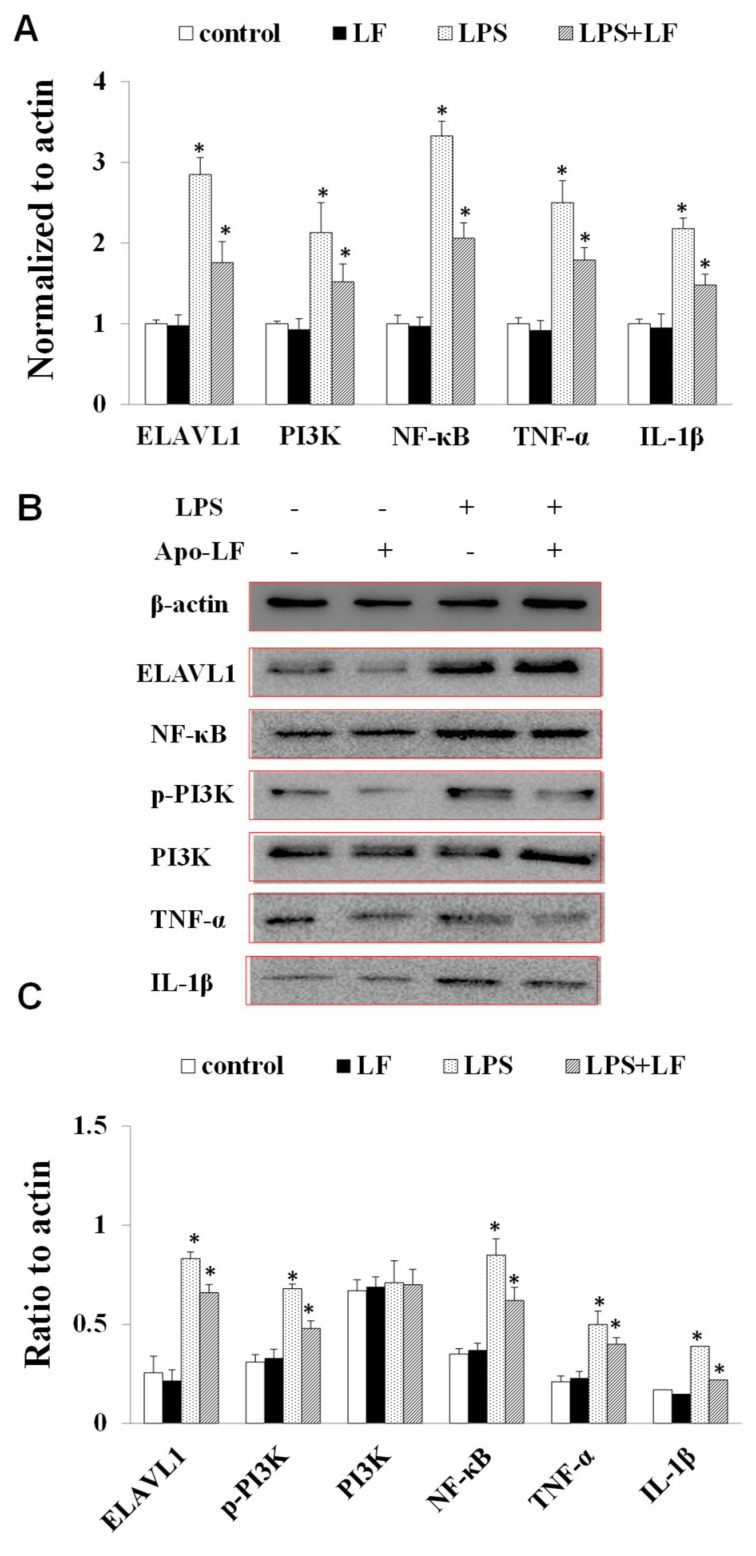
The expression levels of immune-related factors in mouse blood treated by Apo-LF. (**A**) The mRNA expression levels of ELAVL1, PI3K, NF-κB, TNF-α and IL-1β in the blood of mice; (**B**) The protein expression levels of ELAVL1, p-PI3K, PI3K, NF–κB, TNF-α and IL-1β in the blood of mice. *n* = 3 mice per group. Data represent mean ± SD. * *p* < 0.05 by Student’s *t*-test. LF in this figure represents Apo-LF type.

## Data Availability

The immune gene data during the current study are available in the mendeley database (https://doi.org/10.17632/wxxj27ddsy.1, accessed on 10 October 2022).

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
