# Peer review of "Lactoferrin Alleviates Lipopolysaccharide-Induced Infantile Intestinal Immune Barrier Damage by Regulating an ELAVL1-Related Signaling Pathway"

_ijms, 2022, doi:10.3390/ijms232213719_

Round 1

Reviewer 1 Report

ijms-1972742

Lactoferrin Alleviates Lipopolysaccharide-induced Infantile 2 Intestinal Immune Barrier Damage By Regulating 3 ELAVL1-related Signaling Pathway

The article is distinguished by the originality of the selected topic, structured correctly, and written in Standard English. The manuscript presents the role of the innate immune barrier in the intestinal tract, which is important for the overall health of babies and young children. Different bioactivities of lactoferrin (LF) alleviate enteritis and inhibit colon cancer, but the authors try to elucidate the effects of LF on the intestinal immune barrier in infants and young children, relative to the specific mechanism of how LF inhibits infantile enteritis by regulating immune signaling pathways has not been revealed. The pharmacokinetic analyzes of LF were performed in

1) mouse intestinal tissues,

2) gastric tissues and blood, by different administration methods, to confirm the metabolic method of LF in mammals,

 3) constructed in vitro and in vivo models of intestinal immune barrier damage in children using lipopolysaccharide (LPS),

4) the gene and the associated immune pathway were validated at the mRNA and protein levels, the subsets of special immune cells (CD4+ T cells and CD8+ T cells).

 The importance of the study: LF alleviates LPS-induced damage to the intestinal immune barrier in young mice, by regulating ELAVL1-related immune signaling pathways, which would expand the current knowledge of the functions of bioactive proteins in foods within different research layers, as well as benefit preclinical and long-term clinical studies.

The introduction, methods, and results are presented correctly and a logical relationship between them is clearly observed. In an initial review of the article, there were stylistic and grammatical mistakes that I hope the authors will avoid after revision.

Suggest:

1.      Most of the literature used is before 2018, and only 8 % (12 articles) are from the last 3 years. I suggest it be updated! I suggest adding at least 10 more literature sources to validate the results achieved.

2.      The discussion part is small. I suggest that the discussion be expanded further and supported with materials from the last two 21-22 years.

3.      I have no objections to the figures and tables and the graphic design!

Author Response

To Reviewer 1

  The article is distinguished by the originality of the selected topic, structured correctly, and written in Standard English. The manuscript presents the role of the innate immune barrier in the intestinal tract, which is important for the overall health of babies and young children. Different bioactivities of lactoferrin (LF) alleviate enteritis and inhibit colon cancer, but the authors try to elucidate the effects of LF on the intestinal immune barrier in infants and young children, relative to the specific mechanism of how LF inhibits infantile enteritis by regulating immune signaling pathways has not been revealed. The pharmacokinetic analyzes of LF were performed in:

  1) mouse intestinal tissues,

  2) gastric tissues and blood, by different administration methods, to confirm the metabolic method of LF in mammals,

  3) constructed in vitro and in vivo models of intestinal immune barrier damage in children using lipopolysaccharide (LPS),

  4) the gene and the associated immune pathway were validated at the mRNA and protein levels, the subsets of special immune cells (CD4+ T cells and CD8+ T cells).

  The importance of the study: LF alleviates LPS-induced damage to the intestinal immune barrier in young mice, by regulating ELAVL1-related immune signaling pathways, which would expand the current knowledge of the functions of bioactive proteins in foods within different research layers, as well as benefit preclinical and long-term clinical studies.

  The introduction, methods, and results are presented correctly and a logical relationship between them is clearly observed. In an initial review of the article, there were stylistic and grammatical mistakes that I hope the authors will avoid after revision.

Suggest:

  1. Most of the literature used is before 2018, and only 8 % (12 articles) are from the last 3 years. I suggest it be updated! I suggest adding at least 10 more literature sources to validate the results achieved.

  Answer: Thank you so much for your meaningful suggestion! We have searched new literatures from Year 2020 to Year 2022, and added these research results into the Discussion section accordingly.

  1. The discussion part is small. I suggest that the discussion be expanded further and supported with materials from the last two 21-22 years.

  Answer: Yes, based on the new articles, we have added the related research results into the Discussion section, to further expand this part, as follows:

  1. Qin, Z.R.; Zhou, F.F.; Zhang, L. Novel pyroptosis-independent functions of gasdermins. Signal Transduction and Targeted Therapy 2022, 5, 1366-1368.
  2. Lai, Y.J.; Sun, M.; He, Y.; Lei, J.Q.; Han, Y.M.; Wu, Y.Y.; Bai, D.Y.; Guo, Y.M.; Zhang, B. Mycotoxins binder supplementation alleviates aflatoxin B1 toxic effects on the immune response and intestinal barrier function in broilers. Poultry Science 2022, 101(3), 101683.
  3. Chen, H.M.; Wu, X.H.; Xu, C.J.; Lin, J.; Liu, Z.J. Dichotomous roles of neutrophils in modulating pathogenic and repair processes of inflammatory bowel diseases. Precision Clinical Medicine 2021, 4 (4): 246–257. 
  4. Elizabeth, R.; Michael, P.; Jennifer, P.; Raymond, R.; Pinaki, P.; Lewis, R. Probiotic Supplementation in Very Low Birthweight Infants: Effects on Systemic Immunity and Intestinal Inflammation. Current Developments in Nutrition 2022, 6(S1), 706.
  5. Schryvers, A.B. Targeting bacterial transferrin and lactoferrin receptors for vaccines. Trends in Microbiology 2022, 30 (9): 820-830.
  6. Ou, A.T.; Zhang, J.X.; Fang, Y.F.; Wang, R.; Tang, X.P.; Zhao, P.F.; Zhao, Y.G.; Zhang, M.; Huang, Y.Z. Disulfiram-loaded lactoferrin nanoparticles for treating inflammatory diseases. Acta Pharmacologica Sinica 2021, 11: 1913-1920.
  7. Morshedi, V.; Agh, N.; Noori, F.; Jafari, F. A Ghasemi Growth, body composition, physiological responses and expression of immune-related and growth-related genes of Sobaity seabream (Sparidentex hasta) juvenile fed dietary bovine lactoferrin. Iranian Journal of Fisheries Sciences 2020, 19 (6): 3269-3284.
  8. Fan, L.L.; Yao, Q.Q.; Wu, H.M.; Wen, F.; Wang, J.Q.; Li, H.Y.; Zheng, N. Protective effects of recombinant lactoferrin with different iron saturations on enteritis injury in young mice. Journal of Dairy Science 2022, 105, 4791-4803.
  9. El-Nasr, I.A.S.; Mahmou, S.A.; Elnaddar, E.M.; Ammar, H.A. Ferrous Sulphate Alone Versus Combination of Ferrous Sulphate and Lactoferrin for The Treatment of Iron Deficiency Anemia during Pregnancy and Their Effect on Neonatal Iron Store: A Randomized Clinical Trial. The Egyptian Journal of Hospital Medicine 2021, 84, 1955-1960.
  10. Barros, C.A; Sanches, D.; Carvalho, C.A.M.; Santos, R.A.; Souza, T.L.F.; Leite, V.L.M.; Campos, S.P.C.; Oliverira, A.C.; Goncalves, R.B. Influence of iron binding in the structural stability and cellular internalization of bovine lactoferrin. Heliyon 2021, 7 (9), e08087.

  1. I have no objections to the figures and tables and the graphic design!

  Answer: Thank you very much for your suggestions and consideration!

To sum up, we really appreciate your suggestions. Thank you so much for your help and consideration!

Reviewer 2 Report

Li and colleagues describe the role of ‘Lactoferrin Alleviates Lipopolysaccharide-induced Infantile 2 Intestinal Immune Barrier Damage by Regulating 3 ELAVL1-related Signaling Pathway’. The authors presented in this manuscript is well rationalized, executed and interpreted.

Following are the suggestions to the authors that may improve the impact of this manuscript.

1.      Please mention the oral and IP doses in the animal study? How the authors have prepared the formulations for animal study?

2.      Please elaborate the tissue harvesting procedures and how the authors made the calibration set of each tissue?

3.      Is there any recovery experiment was done before tissue analysis? What are the calibration range and LLOQ? Please explain.

4.      Each section of the results needs to have a summary statement to promote readability of the manuscript.

5.      Did the authors calculated the PK parameters? If not please include that as well in the revised manuscript.

6.      Write briefly about the PK parameters in the discussion section.

Author Response

To Reviewer 2

  Li and colleagues describe the role of ‘Lactoferrin Alleviates Lipopolysaccharide-induced Infantile 2 Intestinal Immune Barrier Damage by Regulating 3 ELAVL1-related Signaling Pathway’. The authors presented in this manuscript is well rationalized, executed and interpreted.

  Following are the suggestions to the authors that may improve the impact of this manuscript.

  1. Please mention the oral and IP doses in the animal study? How the authors have prepared the formulations for animal study?

  Answer: We have added the information of oral doses and I.P. doses into the “Methods and Materials” part, as 1 g/kg body weight. And we have addes the description of the formulations preparation into the animal study section, as “At each time point, the intestinal tissues and the stomach tissues were gathered, the tissues (50 mg per mice) were cut into small pieces, 2 mL of normal saline was added into each sample, and the samples were put into the homogenizer to crush the homogenate, once every 30 seconds, and they were homogenized twice, with an interval of 30 seconds between the two times to avoid temperature rise. Then the homogenates were centrifuged at 1500 rpm for 10 min, the supernatant samples were collected for further detection.”.

  1. Please elaborate the tissue harvesting procedures and how the authors made the calibration set of each tissue?

  Answer: Yes, we have elaborated the tissue gathering procedures in “2.2. Animals” section, and added the calibration standards for tissues as “In each mice, the intestinal tissues and the stomach tissues were gathered, and the tissues (50 mg per mice) were cut into small pieces, 2 mL of normal saline was added into each sample...”.

  1. Is there any recovery experiment was done before tissue analysis? What are the calibration range and LLOQ? Please explain.

  Answer: Thank you for your meaningful suggestion. We have carried the recovery experiment with three different concentrations, which are 2000 mg/L (high), 200 mg/L (middle) and 20 mg/L (low). And the recovery rates were in the range of 70%-94%  (presented in the following table). The LLOQ was calculated to 8 mg/L according to standard curve.

Table Recovery rates

No.

Concentration (n=5)

2000 mg/L

200 mg/L

20 mg/L

Sample 1

92.3%

88.2%

70.2%

Sample 2

93.6%

86.4%

74.5%

Sample 3

88.7%

83.9%

78.2%

Sample 4

85.4%

78.3%

84.2%

Sample 5

90.1%

77.2%

81.3%

  1. Each section of the results needs to have a summary statement to promote readability of the manuscript.

  AnswerYes, we have added several summary statements into each sections in “Results” part, to make our expression to be more comprehensible and clear.

  1. Did the authors calculated the PK parameters? If not please include that as well in the revised manuscript.

  AnswerYes, we have calculated the pharmacokinetics parameters, and added them into the revised manuscript, as: “Additionally, we calculated several pharmacokinetics parameters of LF in the present model, as through oral gavage, the Peak Concentrations (Cmax) of LF in blood, intestinal tissue and stomach tissue were 208 mg/L, 1905 mg/L, and 863 mg/L, respectively. Through I.P. injection, the Cmax of LF in blood, intestinal tissue and stomach tissue were 219 mg/L, 23 mg/L and 60 mg/L, respectively. The half life (t1/2) in blood samples was 3.75 h (by oral gavage) and 1.87 h (by I.P. injection), respectively.”

  1. Write briefly about the PK parameters in the discussion section

  AnswerThank you for your suggestion, and we have added description of the PK parameters into the Discussion part, as “Additionally, we performed the pharmacokinetics detection of LF in blood, intestinal tissue and stomach tissue, through oral gavage and I.P. injection, in order to compare the metabolic characteristics of LF in blood, intestines and stomach. We found that the majority of LF could directly reached into intestinal tissue and reserve its primary intact form. As through oral gavage, the Peak Concentrations (Cmax) of LF in intestinal tissue was higher than the ones in blood and stomach tissue, verifying the bioavailability of LF through oral gavage was much higher than the one through I.P. injection. Through comparing the half life (t1/2) of LF in blood samples, we found that 3.75 h (by oral gavage) and 1.87 h (by I.P. injection) were shorter than our expected ones, further suggesting that how to prolong the metabolic course of LF was necessary.”.

To sum up, we really appreciate your suggestions. Thank you very much for your help and consideration!

Round 2

Reviewer 2 Report

No more comments.